# Social Learning versus Individual Learning in the Division of Labour

**DOI:** 10.3390/biology12050740

**Published:** 2023-05-19

**Authors:** Moein Khajehnejad, Julian García, Bernd Meyer

**Affiliations:** Department of Data Science and Artificial Intelligence, Monash University, Clayton, VIC 3168, Australia

**Keywords:** division of labour, social learning, individual learning, evolutionary game theory, adaptive dynamics, cross-learning

## Abstract

**Simple Summary:**

Division of labour is a crucial characteristic of social organisations such as insect colonies and is a key feature in their well-known survival and efficacy. The presence of “laziness”, or inactivity is a widely debated phenomenon that has been observed in some colonies and is puzzling because it goes against the idea that a division of labour would lead to greater efficiency and effectiveness. Inactivity has been previously explained as a by-product of social learning, which is a fundamental type of behavioural adaptation in these colonies. However, this explanation is limited because it is still unclear if social learning governs aspects of colony life. This study explores how inactivity can also emerge similarly from an individual learning paradigm, which is a firmly established paradigm of behaviour learning in insect colonies. Using individual-based simulations backed up by mathematical analysis, the study finds that individual learning can induce the same behavioural patterns as social learning. This is important for understanding the collective behaviour of social insects. The insight that both modes of learning can lead to the same patterns of behaviour opens up new ways of approaching the study of emergent patterns of collective behaviour in a more generalised manner.

**Abstract:**

Division of labour, or the differentiation of the individuals in a collective across tasks, is a fundamental aspect of social organisations, such as social insect colonies. It allows for efficient resource use and improves the chances of survival for the entire collective. The emergence of large inactive groups of individuals in insect colonies sometimes referred to as *laziness*, has been a puzzling and hotly debated division-of-labour phenomenon in recent years that is counter to the intuitive notion of effectiveness. It has previously been shown that inactivity can be explained as a by-product of social learning without the need to invoke an adaptive function. While highlighting an interesting and important possibility, this explanation is limited because it is not yet clear whether the relevant aspects of colony life are governed by social learning. In this paper, we explore the two fundamental types of behavioural adaptation that can lead to a division of labour, *individual learning* and *social learning*. We find that inactivity can just as well emerge from individual learning alone. We compare the behavioural dynamics in various environmental settings under the social and individual learning assumptions, respectively. We present individual-based simulations backed up by analytic theory, focusing on adaptive dynamics for the social paradigm and cross-learning for the individual paradigm. We find that individual learning can induce the same behavioural patterns previously observed for social learning. This is important for the study of the collective behaviour of social insects because individual learning is a firmly established paradigm of behaviour learning in their colonies. Beyond the study of inactivity, in particular, the insight that both modes of learning can lead to the same patterns of behaviour opens new pathways to approach the study of emergent patterns of collective behaviour from a more generalised perspective.

## 1. Introduction

Division of labour is fundamental to the functioning of social organisms and has been central to their study for decades [1]. The separation of tasks among different individuals or groups within a collective allows for the efficient use of resources and increases the chances of survival for the collective as a whole [2,3,4]. Studies have shown that division of labour is prevalent in many socially living organisms, such as ants, bees, termites, and even some mammals [5,6,7,8,9]. Social insect colonies are well known for their intricate organisation and their ability to handle a wide range of tasks simultaneously, including foraging, colony defence, nest construction, temperature regulation, and caring for offspring [6,10]. The colony’s ability to effectively allocate its workforce to these different tasks, adapting to changes in both external conditions and internal needs, is often cited as a key to their ecological success [11,12,13,14]. Understanding the underlying mechanisms of division of labour is fundamental to understanding these social organisations and the emergence of complex social systems in general.

It is well established that both developmental and genetic factors significantly influence the division of labour [15,16]. Additionally, studies have shown that faster and self-organised mechanisms for division of labour exist within colonies, enabling them to rapidly and adaptably respond to shifts in task requirements. Environmental factors or internal shifts within the colony may be responsible for these changes [14,17]. The fast changes in labour division arise from a combination of factors including workforce distribution, interaction structures, and environmental influences [18,19,20,21]. Empirical research has also emphasised the significance of social context and interactions in shaping the task preferences of individuals [22,23]. Individuals generally lack knowledge of the overall state of the colony, thus their behavioural decisions rely on the local information that is readily available to them [24,25]. Interactions among colony members can offer valuable insight into the colony’s condition and act as cues for behaviour, as well as a means for social learning [26,27]. Information acquired through local interactions with other individuals and the environment can often indicate the global state of the colony.

While not frequently discussed in connection to social insects, empirical studies for some species have shown that social learning occurs [26,28,29,30,31]. The fundamental idea of social learning is that an individual observes other individuals and changes their behaviour based on the others’ presence or behaviour. This is a very broad notion. Individuals can be directly influenced by the observed behaviour of other individuals (learning), or they can be influenced by environmental social cues, such as pheromones or simply the presence of others [32,33,34]. Which behaviours are governed by which type of social influence is generally not well understood. In this study, we are only concerned with the dynamics of behaviour learned through imitation, which is already complex in itself and has not yet been widely investigated with mathematical models for social insects. Combining this with other social cues, for example pheromones, is a matter for future extensions of this framework. Thus, in our context, we apply the specific meaning that an individual copies a behaviour that is observed in others. There is indisputable empirical evidence that this happens [26]. Direct interaction or observation is necessary for this to occur. Given the possible complexity of social information exchange, we do not make any assumptions about its underlying mechanisms. We simply posit that individuals are more likely to imitate the behaviours of those who are successful. Independently of this, each individual may explore new behavioural variations with some probability.

We have previously established that certain empirically observed characteristics of colony behaviour, including task specialisation and the emergence of inactive subgroups can arise as a by-product of social learning mechanisms [35]. However, despite the well-established existence of social interactions in colonies, it is uncertain whether these behavioural phenomena can be attributed to mechanisms of social learning with certainty. This is because the exact scope and extent of social learning in insect colonies are not yet well understood. In this study, we investigate whether the same inactivity can also emerge if we only assume individual learning mechanisms.

We juxtapose pure *individual learning* and pure *social learning*, as two distinct methods of information processing by individuals at opposite ends of the spectrum of learning methods. Being based on very different types of information, these learning modes require very different cognitive and sensory capacities. Through a thorough examination of these two extremes, the study aims to gain an understanding of the effect that varying learning assumptions may have on the dynamics of the system.

Our study is motivated by the empirical evidence indicating the presence of both social and individual learning mechanisms in social insects. Bumble bees are a common model system that demonstrate both types of learning. An instance of this is the flower selection behavior in bumblebees, which can be a result of both individual learning and behavior copying [28] and bumblebees exhibit the ability to learn when to use each type of information [36]. We thus need to understand the differences and similarities of these types of learning mechanisms, including their comparative advantages and disadvantages for colony fitness.

To analyse the development of behaviour in a population under the social learning assumption, we employ adaptive dynamics, an analytic approach that originated in Evolutionary Game Theory (EGT) [37,38,39]. Agents follow basic rules to adjust their behaviour in response to an environmental signal, typically referred to as payoff [40]. Adaptive dynamics describes how a group responds to changes by taking into account the actions and interactions of individuals [41]. Evolutionary game theory was initially developed to model changes across evolutionary timescales, where the payoff represents fitness. However, this conceptual framework is not restricted to this timescale and can also be used to model faster processes that involve changes on colony lifetime scales, where payoffs are interpreted as feedback signals instead of fitness [35,42,43]. Our study is explicitly only concerned with these colony lifetime timescales. Interpreted in this way, adaptive dynamics captures how agents modify their task selection by taking into account task performance experience and environmental factors when when working together on multiple tasks.

On the opposite end of the spectrum lies individual learning. It is commonly agreed that individual learning plays a crucial role [3,44,45] for social insects. It enables individuals to adjust to changing environments and improve their task performance over time. This is vital for the colony’s survival, as it allows individuals to adapt to new challenges and make better decisions about how to allocate resources and solve problems. Individuals can adapt their strategies by utilising previously acquired information in their current context [30,46]. The arguably best-established model of task selection in social insects, the reinforcement response threshold model, is centrally based on this notion [47,48,49,50].

In this study, we employ a particularly well-studied form of Reinforcement Learning (RL) [51,52] where agents update their action probabilities using the cross rule of RL [53]. Cross-learning is a relatively simple type of reinforcement learning that is based on individual behaviour and fully aligns with the assumptions of the established adaptive threshold reinforcement model.

We compare the behavioural dynamics under these two learning assumptions for different types of environments. Our central aim is to investigate whether specific types of dynamics can be attributed to a specific learning mechanism, i.e., if they only emerge from social learning but not from individual learning or vice versa. We implement both processes in agent-based models to compare the outcomes. We back up the simulation studies with analytic results derived from adaptive dynamics.

We are specifically interested in an effect previously referred to as *laziness* or inactivity in the population. This refers to the fact that in numerous efficient colonies, a significant portion of the workforce is comprised of inactive workers. This is a frequent occurrence in social organisations, including social insects, animals, humans, etc., which has been observed empirically and explained through modelling studies [35,54,55,56,57,58].

Our results show that identical behavioural dynamics, including the emergence of inactive workers, are observed *independently of the learning mode*. We conclude that this inactivity can be a by-product of the collective learning process in a joint environment but is not conditioned by a particular type of learning.

## 2. Materials and Methods

We commence with a straightforward division of labour problem that only involves the selection of three prototypical tasks, which are labelled X, Y, and Z. To briefly summarise the core of social and individual learning frameworks, we make the assumption that there is a population of agents with a size of *N*, where each agent is entirely characterised by a set of trait values. Each model operates in discrete time steps. In each step, agents engage in group interactions of a predetermined size of *n* (known as “*n*-player games” in game theory terminology), and the population consists of *K* distinct *n*-player games, $G_{1},G_{2},\cdots ,G_{K}$, where $K=\frac{N}{n}$. Agent *i* in game $G_{k}$ obtains a payoff $\Pi_{i,G_{k}}$ from the group interaction, which is typically influenced by both the trait values of agent *i* and those of the other agents participating in the game. Nonetheless, the mechanism for learning (updating rule) varies depending on the type of learning. With *social learning*, agents acquire knowledge from one another by imitating or adopting the traits of another agent. Note that this can be viewed as being influenced by recruitment and imitation. In the event that the recruitment effort is adjusted based on task performance experience, proficient agents are more likely to be imitated. On the other hand, in *individual learning*, instead of imitation, each individual exploits their own experience and reinforces the probability of engaging in a certain task when the individual engages in it successfully. Figure 1 illustrates the general schematic of the dynamic process of both mechanisms.

### 2.1. Social Learning Setting

To study the transition of behaviour in a population under social learning assumptions, we use adaptive dynamics as a framework of evolutionary game theory. Formally, agent *i* is characterised by a triple $\left(x_{i},y_{i},z_{i}\right)$, where the trait values *x*, *y*, and *z* can be interpreted as the average fraction of effort invested into the first, second, and third task, respectively. As $\forall i:\;x_{i}+y_{i}+z_{i}=1$, we can model a population of *N* workers as a two-dimensional vector of trait values ${\left(x_{j},y_{j}\right)}_{j=1,\cdots ,N}$. As we are predominantly interested in the emergence of inactivity, we model inactivity as a third “pseudo-task” that does not generate benefit and has no cost (see [35]). We thus have to have two “normal” tasks (*X* and *Y*) with collective benefits and inactivity as a third “pseudo” task (*Z*).

In numerous social organisations, such as social insects, the benefits arising from task completion are shared and are contingent on the cumulative effort invested, rather than solely on individual effort. An essential characteristic that we intend to investigate is task combinations in which a suitable number of workers need to perform multiple tasks to ensure the smooth functioning of the colony. Examples include brood care and thermoregulation. We model this with a multiplicative coupling of benefit $B_{X}$ of Task *X* and $B_{Y}$ of Task *Y* as follows:(1)$$\begin{array}{c} B\left(X_{k},Y_{k}\right)=\frac{1}{n}\;B_{X}\left(\sum_{x\in X_{k}}x\right)\cdot B_{Y}\left(\sum_{y\in Y_{k}}y\right). \end{array}$$
where $X_{k}$, $Y_{k}$ are the collective engagement levels of all individuals in $G_{k}$. The direct and immediate cost of executing a task, on the other hand, is borne by the agent performing the task and depends on the individual effort invested. Here, the third task (*Z*; the level of inactivity) is assumed to cause no cost for the individuals. Hence, costs for multiple tasks are additive as below:(2)$$\begin{array}{c} C\left(x_{j},y_{j}\right)=C_{X}\left(x_{j}\right)+C_{Y}\left(y_{j}\right). \end{array}$$

The payoff for individual *j* participating in game $G_{k}$ is given as the difference between the benefit obtained and the cost incurred. In game $G_{k}$, individual *j* thus receives a payoff $\Pi_{j,G_{k}}$ as:(3)$$\begin{array}{c} \Pi_{j,G_{k}}=\;B\left(X_{k},Y_{k}\right)-C\left(x_{j},y_{j}\right). \end{array}$$

The shape of the cost and benefit functions reflect the properties of the tasks and the environment. Details are given in Appendix A. (We follow [35]: *X* is a task with a concave benefit shape and marginally decreasing cost such as thermoregulation tasks in an ant colony; *Y* is a task with sigmoidal (thresholding) benefit shape and marginally decreasing costs such as brood care or defence tasks; and *Z* indicates inactivity (forgone effort), which produces no benefit and bears no cost.) Appendix B also explains the theoretical analysis and updating rules of adaptive dynamics. More details on the update rule associated with the Cross-learning are in Appendix C.

### 2.2. Individual Learning Setting

We analyse reinforcement learning, which is considered one of the simplest and most widely studied forms of individual or experience-based learning models. Reinforcement learning is a process in which an agent modifies its internal mixed strategy, which represents its behavioural disposition, modelled by a set of probability distributions determining how individual actions are selected. If a task execution results in a high payoff in the past, its future probability increases, reinforcing the behaviour associated with the action. (This is akin to lowering the threshold in the reinforced threshold model). Reinforcement protocols have substantial empirical support and have been widely used to model a range of complex behaviours in social and biological systems [59]. We study RL in a population game as an appropriate representation of collective behaviour modification in a colony.

We employ a particular form of RL where agents update their action probabilities using the cross rule of RL [53]. Cross-learning is a straightforward form of individual-based reinforcement learning that aligns with the widely accepted threshold reinforcement model for the division of labour in social insects.

For the cross-learning framework, the most intuitive choice would be to characterise each worker by the probability with which she engages in a particular task. Agent *i* would thus be represented by a triple $\left(\pi_{i,X},\pi_{i,Y},\pi_{i,Z}\right)$ where $\pi_{i,X}$, $\pi_{i,Y}$, and $\pi_{i,Z}\in R$ are the probabilities of executing the first, second, and third tasks, respectively ($\forall i:\;\pi_{i,X}+\pi_{i,Y}+\pi_{i,Z}=1$).

Note that this would imply an important difference in modelling the social learning and the individual learning processes. Instead of dividing the invested effort between three tasks (as in the social learning paradigm), each individual fully engages in a single task with a probability given by its trait values. Thereby, given the conventional form of cross-learning, we can only account for discrete task engagement patterns for individuals. However, to account for the possibility of non-binary participation in tasks that involve continuous trait values, which are commonly used in social learning, we extend the conventional cross-learning algorithm. Instead of having a binary choice for selecting a task or not, we model the level of engagement by discretising the levels of engagement into bins. Each bin corresponds to a pair of ranges for both *x* and *y* traits, model the proportion of effort invested in the tasks, exactly as in the social learning paradigm, and each individual bin is assigned a probability of being selected. One might argue that the level of engagement (in social learning) could be interpreted as the long-term average of the task execution frequency. While this is a reasonable interpretation, this will only result in comparable payoffs if the expectation of the payoff of individual task engagements is identical to the payoff of the expected level of engagement. This is generally not the case for non-linear payoff functions.Here, we choose bins of size 0.05 × 0.05 resulting in 210 different pairs of value ranges for *x* and *y* traits.

Figure 2 illustrates the binarising process in the proposed modified version of the cross-learning algorithm.

Then, for the purpose of comparison, we can define the multiplicative coupling of benefit $B_{X}$ of Task *X* and $B_{Y}$ of Task *Y* and immediate costs of $C_{X}$ and $C_{Y}$ similar to the previous setting in Section 2.1.

More details on the update rule associated with the adaptive dynamics can be found in Appendix B.

## 3. Results

We compared the behavioural trajectories of the models discussed above, dependent on parameters $b_{1}$, $b_{2}$, *w*, and β, which are embedded in the benefit and cost functions and reflect the properties of the environment (see Table 1). We implemented the models as individual-based, discrete-time simulations starting from a monomorphic population. At each time step, the population was randomly divided into *K* sets of *n* individuals each, where *n* is a fixed group size. Each individual received a payoff determined by their trait values and the composition of the group they are part of. In the case of social learning, the individuals were then recruited to successful behaviours (technically, individuals imitate the trait values of others with a probability that is determined by the recruiter’s performance compared to the average performance of the entire population). Each trait value could also undergo a slight change that could be considered an autonomous exploration of behaviour through variation, similar to a mutation. In individual learning, due to the update rules related to cross-learning, each agent modifies their trait values at each time step by reinforcing the probability of the action executed according to the task-related reward experienced. Full details of each method are given in Algorithms 1 and 2.

Figure 3 shows the simulation results of both models for different sets of environmental parameters. (The source code of the simulations is available at this GitHub link).

The different behaviour variations were classified into three groups:

**Fully specialised:** Regardless of the boundary conditions, the entire population uniformly shifts towards full engagement in a single task (task *Z* or inactivity), resulting in inviability.

**Branching:** After initial movement toward a shared level of engagement, the population splits into two (or more) co-existing traits. These sub-populations show different levels of engagement in the three tasks.

**Uniform behaviour; fully generalised:** In this case, all individuals move toward a shared level of engagement in each of the three tasks (i.e., a shared set of all trait values with a certain level of inactivity). From the EGT perspective, this represents an Evolutionary Stable Strategy (ESS).

The simulation results in Figure 3 illustrate that both learning paradigms resulted in the same behaviour in all behavioural environments. Given a monomorphic initial population, all individuals first move towards a fixed point (red dot in the streamline plots) starting from an initial set of trait values (green dot in the streamline plots). This fixed point was predicted analytically using adaptive dynamics as shown in the streamline plots of Figure 3. In certain environments, a branching behaviour occurs after reaching the fixed point. This split is also predicted by adaptive dynamics for the case of social learning. The simulations show that this is not unique to social learning but that both individual and social learning dynamics exhibit the same split (see Left and Right simulation results in Figure 3b for individual learning and social learning results respectively). More intriguingly, the results depict that both learning mechanisms can simulate the emergence of inactivity (i.e., non-zero engagement in task *Z* at the steady state) in certain parameter ranges. Thus, the models suggest that under certain environmental conditions, inactivity can arise simply as a by-product of the collective adjustment process in a joint environment without being restricted to a specific form of learning.
**Algorithm 1** Social Learning: At each generation *t*, each individual *j* updates its strategy (${x^{t}}_{j},{y^{t}}_{j}$) following an imitation phase and a mutation with probability μ **Require:** A population of size *N* with a strategy profile $p^{0}=\left\{\left(x_{1}^{0},y_{1}^{0}\right),\left(x_{2}^{0},y_{2}^{0}\right),\cdots ,\left(x_{N}^{0},y_{N}^{0}\right)\right\}$ at generation t=0; $\left(x_{j}^{0},y_{j}^{0}\right)=\left(x_{0},y_{0}\right)\in \left(\left[0,1\right],\left[0,1\right]\right)$; selection intensity, ζ; mutation rate, μ; standard deviation for Gaussian mutations, σ.
1: P←[]2: **for** t=1:T **do**3:    $P.append(p^{t-1})$4:    $\mathrm{generate}\;\;K=\frac{N}{n}\;\;\mathrm{random}\;\mathrm{games}$5:    **for** j=1:N **do**6:      $\Pi_{j,G_{k}}\leftarrow B\left(X_{k}^{t-1},Y_{k}^{t-1}\right)-C\left(x_{j}^{t-1},y_{j}^{t-1}\right)$7:      where   $j\in G_{k};\;\;k\in \left\{1,\cdots ,K\right\}$8:    **end for**9:    **for** j=1:N **do**10:      $\mathrm{imitates}\;\;l\;\;\mathrm{with}\;\mathrm{probability}\;\;\frac{e^{\zeta \Pi_{l}}}{\sum_{m=1}^{N}e^{\zeta \Pi_{m}}}$11:      $x_{j}^{t}\leftarrow x_{l}^{t-1}$12:      $y_{j}^{t}\leftarrow y_{l}^{t-1}$13:    **end for**14:    **for** j=1:N **do**15:      **if** random()<μ **then**16:         $x_{j}^{t}\leftarrow max\left(0,min\left(N\left(x_{j}^{t},\sigma \right),1\right)\right)$17:         $y_{j}^{t}\leftarrow max\left(0,min\left(N\left(y_{j}^{t},\sigma \right),1\right)\right)$18:      **end if**19:    **end for**20:    $p^{t}\leftarrow \left\{\left(x_{1}^{t},y_{1}^{t}\right),\left(x_{2}^{t},y_{2}^{t}\right),\cdots ,\left(x_{N}^{t},y_{N}^{t}\right)\right\}$21: **end for**22: **Return** $P=\{p^{0},p^{1},\cdots ,p^{T-1}\}$


**Algorithm 2** Modified cross-learning: At each generation *t*, each individual *j* updates its strategy (${x^{t}}_{j},{y^{t}}_{j}$) and probability distribution ($\pi_{j,1}^{t},\cdots ,\pi_{j,m}^{t}$) over *m* possible action bins $B_{1},\cdots ,B_{m}$ with learning rate of α **Require:** A population of size *N* with a strategy profile $p^{0}=\left\{\left(x_{1}^{0},y_{1}^{0}\right),\left(x_{2}^{0},y_{2}^{0}\right),\cdots ,\left(x_{N}^{0},y_{N}^{0}\right)\right\}$ and probability distribution profile $\Pi^{0}=\left\{\left(\pi_{1,1}^{0},\cdots ,\pi_{1,m}^{0}\right),\cdots ,\left(\pi_{N,1}^{0},\cdots ,\pi_{N,m}^{0}\right)\right\}$ at generation t=0; $\left(x_{j}^{0},y_{j}^{0}\right)=\left(x_{0},y_{0}\right)\in B_{l}$; $\pi_{j,l}^{0}=1,\pi_{j,a}^{0}=0\;\;\;\forall a\neq l$; learning rate, α.
1: P←[]2: **for** t=1:T **do**3:    $P.append(p^{t-1})$4:    $\mathrm{generate}\;\;K=\frac{N}{n}\;\;\mathrm{random}\;\mathrm{games}$5:    **for** j=1:N **do**6:      agentjchoosesanactionbin7:      $a_{j}^{t-1}\in \left\{B_{1},\cdots ,B_{m}\right\}$8:      $\left(x_{a_{j}^{t-1}},y_{a_{j}^{t-1}}\right)\leftarrow \mathrm{center}\;\mathrm{point}\;\mathrm{of}\;\;a_{j}^{t-1}$9:      $x_{j}^{t-1}\leftarrow max\left(0,min\left(N\left(x_{a_{j}^{t-1}},\sigma \right),1\right)\right)$10:      $y_{j}^{t-1}\leftarrow max\left(0,min\left(N\left(y_{a_{j}^{t-1}},\sigma \right),1\right)\right)$11:      $I_{j}=l\;\;\;\;\;\;s.t.\;\;\left(x_{j}^{t-1},y_{j}^{t-1}\right)\in B_{l},\;\;l\in \left\{1,\cdots ,m\right\}$12:    **end for**13:    **for** j=1:N **do**14:      $\Pi_{j,G_{k}}\leftarrow B\left(X_{k}^{t-1},Y_{k}^{t-1}\right)-C\left(x_{j}^{t-1},y_{j}^{t-1}\right)$15:      where   $j\in G_{k};\;\;k\in \left\{1,\cdots ,K\right\}$16:    **end for**17:    **for** j=1:N **do**18:      $\pi_{j,l}^{t}\leftarrow \pi_{j,l}^{t-1}+\left\{\begin{array}{ll} \alpha (\Pi_{j,G_{k}}-\Pi_{j,G_{k}}\cdot \pi_{j,l}^{t-1}) & l=I_{j}\\ \alpha \left(-\Pi_{j,G_{k}}\cdot \pi_{j,l}^{t-1}\right) & o.w \end{array}\right.$19:    **end for**20:    $p^{t}\leftarrow \left\{\left(x_{1}^{t},y_{1}^{t}\right),\left(x_{2}^{t},y_{2}^{t}\right),\cdots ,\left(x_{N}^{t},y_{N}^{t}\right)\right\}$21: **end for**22: **Return** $P=\{p^{0},p^{1},\cdots ,p^{T-1}\}$


Finally, we repeat our analysis in the same environmental setting as the branching region in Figure 3b using the modified cross-learning framework but with a larger bin size of 0.2. This divides the entire space of possible *x* and *y* paired value ranges to six possible bins. We coarse-grain the model in order to address the concern that a fine-grained bin model is surely not biologically plausible. What is plausible, though, is that the individual may have some concept of executing a task “always”, “never”, “frequently”, or “infrequently”. This is adequately captured in the coarse-grained bin model. As expected, the results in Figure 4 show qualitatively the same behaviour as in Figure 3 but with noise added. These findings confirm the fact that the findings do not change under a cognitively plausible model of trait values.

## 4. Discussion

Division of labour is essential for the survival and ecological success of social organisations. By dividing tasks among individuals, a social organisation can ensure that the most skilled or efficient individuals are performing specific tasks, which can lead to increased productivity and overall success. Dividing labour can also promote specialisation in the population, as individuals are able to focus on specific tasks and develop expertise in those areas. Furthermore, it also allows for flexibility and the ability to respond quickly to changes in the environment and internal requirements.

Social insect colonies are examples of the most ecologically successful life forms, and an efficient division of labour is a critical aspect of their success. There has been a significant amount of research on the division of labour in social insects [1]. However, much of this research has focused on the impact of internal factors such as genetics [60], morphology [61], and hormones [62]. In comparison, there has been relatively less focus on the impact of the environment on task choices at the individual level and the underlying mechanisms of social interactions and their role in regulating the division of labour.

We studied two of the most widely used methods for modelling the mechanisms of the division of labour in social organisations: social learning and individual learning. Very few previous studies have focused on comparing the similarities and differences in the outcomes resulting from the different update rules used. A comprehensive comparison of the two frameworks in various environmental settings is crucial in understanding the advantages and limitations of each assumption. It will help in the better understanding of the underlying mechanisms of the division of labour, in general, and more specific phenomena such as the emergence of inactivity as observed in empirical data, in particular.

In this study, we have attempted to gain a deeper understanding of the implications of these two different learning paradigms for a specific behavioural phenomenon observed in social colonies: the emergence of inactive subgroups that do not participate in the collective action that sustains the colony.

Previous studies have posited that this particular phenomenon, an instance of branching behaviour, is a by-product of the learning mechanism [35]. However, it was unclear whether these aspects of dynamics were indeed influenced by the presence of social interactions and the learning mechanism itself. By comparing and contrasting the results of both individual and social learning paradigms across different environmental conditions, we found equivalent behavioural outcomes in both cases. Specifically, we demonstrated that an individual, experience-based learning approach can also lead to inactivity in the population. This supports the hypothesis that regardless of the dominant learning mechanism in the colony, this aspect of colony life can arise as an artefact of the collective behaviour modification in a joint environment but is not necessarily restricted to a specific learning mechanism.

Using mathematical intuition, this is not entirely surprising. There are deep correspondences between cross-learning and social learning. The seminal contribution by Borgers [63] first established that cross-learning and replicator dynamics (Appendix D), a formal model of learning by imitation, exhibit similar dynamics. However, the important restriction of this result is that it only applies to the learning of a single individual and that it can only be proven *in expectation* and in the continuous-time limit. The setting of a social insect colony, however, is necessarily population-based learning. The term population-learning is adapted from the reinforcement literature and can refer to any interacting collective, rather than to the specific meaning of population in biology. Börgers and Sarin’s finding was later extended by Lakhar and Seymour to show that population-based cross-learning evolves according to a specific form of the replicator equation under certain conditions, the so-called replicator continuity equation [64]. This equation is a partial differential equation that describes the changes in the population state over time. This was a very important finding, but it is restricted to replicator dynamics, which can only capture discrete behavioural states.

Adaptive dynamics, which we have used here, can capture continuous behaviour parameters (such as an engagement level) and can analytically predict whether a population will split into subgroups. What we have demonstrated in this paper is that adaptive dynamics and population-based cross-learning exhibit qualitatively equivalent dynamics in the context of the study of inactivity.

The presence of the studied inactive subgroups is a common occurrence in collective behavior, observed in organisations such as active particles, insects, animals, and humans [55,65]. Social insect colonies, in particular, can have over half of their workers inactive at a given time [66], which is surprising given the low individual selfishness levels in these colonies [13]. Despite our hypothesis that inactivity can arise as a by-product of the task allocation process and independent of the learning mode, in certain situations, this inactivity has been proposed to have a functional purpose. The main hypothesis for the functional role of inactive workers is that they serve as a reserve workforce that can be mobilised quickly when there is a sudden loss of workers or unexpectedly high task demands, thereby increasing colony flexibility and resilience [67,68]. Nevertheless, the benefits derived from having a reserve workforce of inactive individuals have never been quantified, either empirically or otherwise, leading many empirical research to still question this hypothesis [69]. Examples of other explanations include sleep or rest time of individuals [70,71], or delays occurring during task switching and the time the workers require to asses the collected information about task demands without engaging in any work [72]. However, the variation among individuals in social insect colonies in terms of their amount of inactivity cannot be fully explained by the need for resting periods alone [13] and although challenging, the purposeful activity of searching for a task, or “patrolling” should be distinguished from aimless and inactive wandering [73]. This suggests the necessity for additional research on the subject, both through empirical studies and theoretical or modeling work, as proposed in this study.

Perhaps the main limitation of our study is that we have solely looked at social learning and individual learning in isolation. Yet, it is likely that, in many circumstances, both may occur simultaneously and even intermingle, possibly for individual task selection or even in a context-dependent manner, as shown in previous research [36]. While we have focused on analyzing isolated forms of each learning paradigm, investigating such mixed modes is a task for future studies. Our paper’s primary objective was to demonstrate that the overall behavior dynamics in the population are similar under both learning paradigms. Therefore, it is probable that a combination of learning mechanisms would also manifest similar dynamics. It should be relatively straightforward to confirm this with simulations, however a mathematical framework that captures both modes simultaneously is unclear.

To conclude, we believe that our approach, beyond the study of this particular phenomenon of collective behaviour (i.e., inactivity), hints at new pathways for studying collective behaviour in animal groups from a generalised perspective without having to assume (or know) a restrictive model of learning. To explore this possibility further, the exact conditions under which these equivalences hold will have to be established formally and we hope to do so in future work.

## Figures and Tables

**Figure 1 biology-12-00740-f001:**
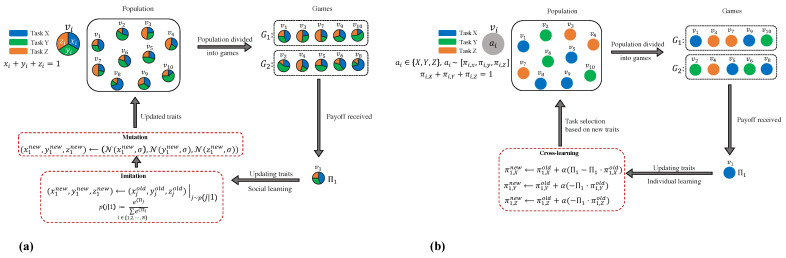
(**a**) A diagrammatic representation of various stages in the social learning paradigm from an EGT perspective using a sample population with size N=10. Each circle symbolises an individual in the population, and its segmentation into three colors denotes the distribution of involvement in the three tasks or response traits: *x*, *y*, and *z*. The various steps of the process are demonstrated for an example agent, $v_{1}$, in the population. (**b**) Schematic diagram of different steps in the individual learning paradigm given three possible task choices. Each circle represents an individual and each colour represents the selected task by that individual. The different steps of the process are illustrated for a sample agent, $v_{1}$, in the population.

**Figure 2 biology-12-00740-f002:**
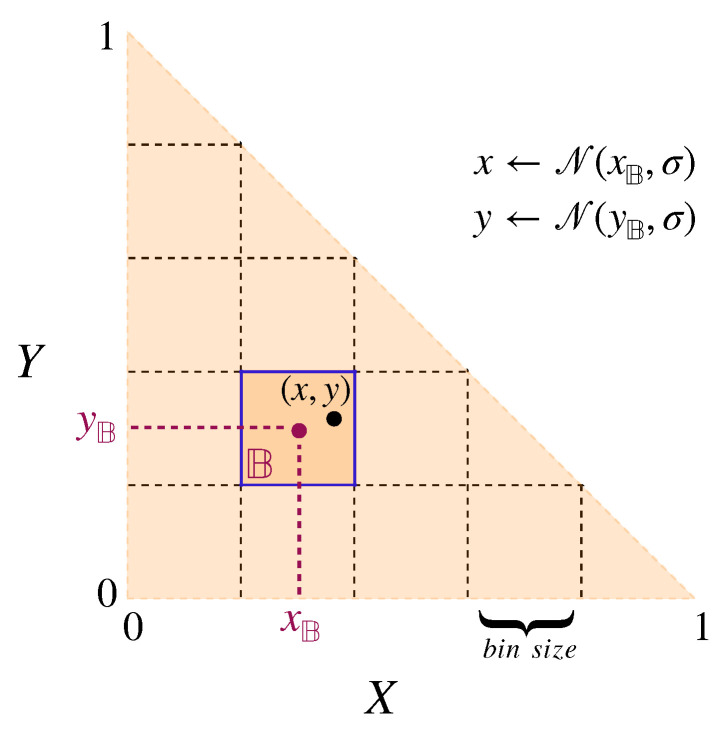
Illustrating the process of binarising the space of possible (x,y) choices in the modified cross-learning algorithm. At each step of updating the trait value pair, (x,y), a single bin B is first selected. The new (x,y) pair is then calculated, applying a random Gaussian distribution around the center point of the bin, which is shown as $\left(x_{B},y_{B}\right)$.

**Figure 3 biology-12-00740-f003:**
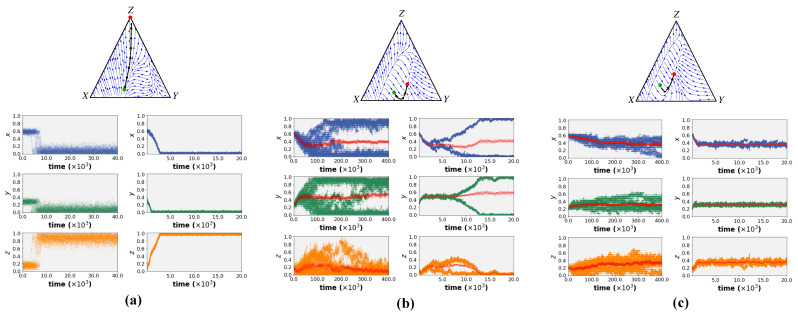
Streamline plots, as well as **Left:** individual learning, and **Right:** social learning simulation results for different environmental settings. The streamline depicts the development of a monomorphic population from the initial point (in green) to the fixed point (in red). (**a**) **Fully specialised:** $b_{1}=16$, $b_{2}=-6$, ω=0.2. (**b**) **Branching:** $b_{1}=20$, $b_{2}=-4$, ω=0.3. (**c**) **Uniform behaviour; fully generalised:** $b_{1}=28$, $b_{2}=-6$, ω=0.5. The average population trait values are plotted in red which closely match between individual learning (cross-learning simulations) and social learning (adaptive dynamics simulations).

**Figure 4 biology-12-00740-f004:**
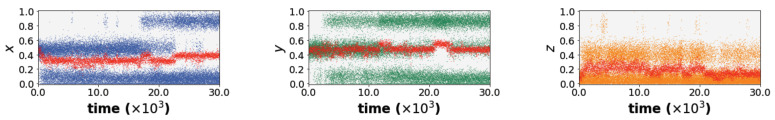
Individual learning simulation results with bin sizes equal to 0.2 in the **branching** region: $b_{1}=20$, $b_{2}=-4$, ω=0.3. This binarisation results in 6 different pairs of value ranges for x and y traits; the settings mimic a case of 6 different tasks in an unmodified cross-learning framework.

**Table 1 biology-12-00740-t001:** Model parameters.

Notation	Definition	Interpretation	Values
$b_{1}$	linear benefit coefficient for task *X* (e.g., thermoregulation)	efficiency of regulation	{16,20,28}
$b_{2}$	quadratic benefit coefficient for task *X*	shape of benefit for task *X* (homeostatic vs. maximising)	{−4,−6}
β	slope of benefit for task *Y* (e.g., brood care)	larger values indicate higher efficiency per unit of work	3
*w*	inflection point of benefit for task *Y*	1/w indicates minimum amount of work required	{0.2,0.3,0.5}
*n*	group size for individual interactions		5
ζ	selection intensity in social learning		2
μ	mutation rate in social learning	probability of behaviour exploration	0.01
σ	mutation size in social learning	amount of behaviour variation	0.005
α	learning rate (step size) in individual learning	determines to what extent newly acquired information overrides old information	0.01

## Data Availability

The scripts and simulations supporting the reported results can be found in https://github.com/Moein-Khajehnejad/Social_learning-vs.-Individual_learning.

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
