# Peer review of "Social Learning versus Individual Learning in the Division of Labour"

_biology, 2023, doi:10.3390/biology12050740_

Round 1

Reviewer 1 Report

Overall, the article is well-developed. The study is ambitious, with many frameworks aimed to be explored. However, certain elements need to be improved. The manuscript could be improved in providing more details on data analysis, discussion, and conclusions. In its current form, the manuscript needs some improvements based on the following comments and suggestions:

1. The introduction is quite helpful and provides context for the subject matter of this study. The study objectives and goals, as well as their background, should be clearly described in the manuscript. 

2. The research advances knowledge while challenging preexisting paradigms. The Introduction of the study presents a few pertinent and intriguing concepts. 

3. This manuscript makes intriguing analyses and introduces fresh topics for discussion. Only a small handful of sources are recent and up to date.

4. The data analysis techniques are adequate, however the authors should provide additional information on them. The research employs techniques that are well-known and often used. The methodologies may have been improved upon in light of the research setting in order to generate insightful perspectives for future investigations. The methods have generally undergone validation, which the authors explain in detail.

5. The writers use the findings to offer insightful explanations. More work should be put into providing analogous perspectives with earlier studies, though. The conclusions of this empirical study could be strengthened by the addition of new details and suggestions for future investigation. The theoretical contributions of the study are just briefly mentioned by the authors. Additional details might be helpful. The study's limitations ought to be made more explicit. 

6. Further details and suggestions for future research could be added to the conclusions of this empirical paper. Based on the results of the work, just a few suggestions for further study are made. 

7. The theoretical contributions are briefly mentioned. 

8. The restrictions are left unmentioned. The outcomes are not contrasted with those from additional sources or studies.

All the best to the Authors in their future academic endeavours!

Author Response

We thank the reviewer for their detailed comments and constructive feedback, which has led to a stronger and more comprehensive revised manuscript.

Overall, the article is well-developed. The study is ambitious, with many frameworks aimed to be explored. However, certain elements need to be improved. The manuscript could be improved in providing more details on data analysis, discussion, and conclusions. In its current form, the manuscript needs some improvements based on the following comments and suggestions:

  1. The introduction is quite helpful and provides context for the subject matter of this study. The study objectives and goals, as well as their background, should be clearly described in the manuscript. 

The authors would like to thank the reviewer for all the raised points and comments. In order to address the above point, we have added a paragraph to the ‘Introduction” section (Lines 97-104 of the revised manuscript) to further clarify the motivations and objectives of the current study.

  1. The research advances knowledge while challenging preexisting paradigms. The Introduction of the study presents a few pertinent and intriguing concepts. 
  2. This manuscript makes intriguing analyses and introduces fresh topics for discussion. Only a small handful of sources are recent and up to date.

The authors appreciate the detailed points brought up by the reviewer and the encouraging comments.

  1. The data analysis techniques are adequate, however the authors should provide additional information on them. The research employs techniques that are well-known and often used. The methodologies may have been improved upon in light of the research setting in order to generate insightful perspectives for future investigations. The methods have generally undergone validation, which the authors explain in detail.

We have made sure that the details and additional information about the mathematical analyses are included in the Appendix attached to the manuscript.

  1. The writers use the findings to offer insightful explanations. More work should be put into providing analogous perspectives with earlier studies, though. The conclusions of this empirical study could be strengthened by the addition of new details and suggestions for future investigation. The theoretical contributions of the study are just briefly mentioned by the authors. Additional details might be helpful. The study's limitations ought to be made more explicit. 
  2. Further details and suggestions for future research could be added to the conclusions of this empirical paper. Based on the results of the work, just a few suggestions for further study are made. 
  3. The theoretical contributions are briefly mentioned. 
  4. The restrictions are left unmentioned. The outcomes are not contrasted with those from additional sources or studies.

We have now added new text to highlight the main limitations of the study [lines 360-369], contrast the findings to some previous empirical sources [lines 339-359], as well as the most important points for future work [lines 68-76].

Reviewer 2 Report

In this paper,  division of labour is investigated through the lens of two fundamental types of behavioural adaptation, namely individual learning  and social learning. The behavioral dynamics are examined in diverse environmental conditions under the social and individual learning assumptions, respectively. It was concluded that individual learning can elicit the same behavioral patterns as social learning.

The investigation of the underlying mechanisms of the division of labour is an important topic. Indeed, it is important to clarify whether these aspects of dynamics are influenced by the presence of social interactions or the learning mechanism itself. A comparison of the two frameworks in various environmental settings is comprehensive and provides readers with a good understanding of the advantages and limitations of each assumption. Methods are well-described. Figures and tables are useful for the readers.

 I believe that the paper is original and well-structed and it can be published even in its current form.

Author Response

In this paper,  division of labour is investigated through the lens of two fundamental types of behavioural adaptation, namely individual learning  and social learning. The behavioral dynamics are examined in diverse environmental conditions under the social and individual learning assumptions, respectively. It was concluded that individual learning can elicit the same behavioral patterns as social learning.

The investigation of the underlying mechanisms of the division of labour is an important topic. Indeed, it is important to clarify whether these aspects of dynamics are influenced by the presence of social interactions or the learning mechanism itself. A comparison of the two frameworks in various environmental settings is comprehensive and provides readers with a good understanding of the advantages and limitations of each assumption. Methods are well-described. Figures and tables are useful for the readers. I believe that the paper is original and well-structed and it can be published even in its current form.

The authors appreciate the detailed points brought up by the reviewer and the encouraging comments. The introduction of the revised manuscript is now strengthened by including more details about the empirical backgrounds and the motivations behind this study [lines 97-104 and 68-76].

Reviewer 3 Report

The manuscript by Khajehnejad et al. studied the mechanism of inactivity (laziness) in the collective behavior in social colonies using mathematical modeling, and the results supported that the emergence of inactivity can be induced by social or individual learning alone. This is an interesting topic and the manuscript is well written. I don’t have the expertise to evaluate the algorithms in the study, but there are a few questions/concerns that I hope the authors can address before publication.

Lines 29-31: “Social insect colonies are …and care for offspring”. Please add references to back up the statement.

Line 37: Please change to “developmental and genetic factors”. Morphological factor is not always involved in division of labor, for example, in age polyethism.

Lines 54-55 and 127-130: In this study, social learning is defined by one individual imitating a behavior performed by another. The collective behavior in social colonies is mediated through responses of individuals to local cues, meaning that there is always a cue (from nestmates or the environment). So, how does this study distinguish between learned behavior and innate response to the same social cue? In many social behaviors, including recruitment via trail pheromones and defense via alarm pheromones, individuals exhibit the same behavior not because one imitates another (learning), but because they all sense and respond to the same cue (innate behavior). Some clarification may be needed.

Line 132: Please change “she” to “the individual”, as there are also male workers in termite colonies.

Discussion: The paper mentions that inactivity is a “by-product” of learning, and such a statement implies this behavior does not have any adaptive value. However, in other publications, there are alternative hypotheses and non-learning mechanisms examined and discussed, and some suggest “laziness” may play a functional role. For example, the lazy workers might be a reserve labor force in the colony, they might be younger and physiologically less efficient (not necessarily related to learning, but rather due to developmental plasticity mediated by social stimuli), or they might be selfishly reproducing or developing ovaries. It would be nice to introduce and discuss the non-learning mechanisms and potential functions of inactive behavior in the manuscript.

Author Response

We thank the reviewer for their detailed comments and constructive feedback, which has led to a stronger and more comprehensive revised manuscript.

The manuscript by Khajehnejad et al. studied the mechanism of inactivity (laziness) in the collective behavior in social colonies using mathematical modeling, and the results supported that the emergence of inactivity can be induced by social or individual learning alone. This is an interesting topic and the manuscript is well written. I don’t have the expertise to evaluate the algorithms in the study, but there are a few questions/concerns that I hope the authors can address before publication.

  1. Lines 29-31: “Social insect colonies are …and care for offspring”. Please add references to back up the statement.

We thank the reviewer for bringing this point to our attention. Appropriate references were added to this sentence in the revised version of the manuscript.

  1. Line 37: Please change to “developmental and genetic factors”. Morphological factor is not always involved in division of labor, for example, in age polyethism.

We thank the reviewer for bringing this fine point to our attention. The phrasing has been corrected as suggested.

  1. Lines 54-55 and 127-130: In this study, social learning is defined by one individual imitating a behavior performed by another. The collective behavior in social colonies is mediated through responses of individuals to local cues, meaning that there is always a cue (from nestmates or the environment). So, how does this study distinguish between learned behavior and innate response to the same social cue? In many social behaviors, including recruitment via trail pheromones and defense via alarm pheromones, individuals exhibit the same behavior not because one imitates another (learning), but because they all sense and respond to the same cue (innate behavior). Some clarification may be needed.

We agree that social learning can indeed occur in various forms and be driven by various influences, including both influences from the environment and direct behaviour imitations. Which behaviours are governed by which type of social influence is generally not well understood. Our study only addresses the latter kind, i.e. direct imitation of behaviour. It is well known from experiment studies that this type of social learning exists in social insect colonies [for example, Worden, Bradley D., and Daniel R. Papaj. "Flower choice copying in bumblebees." Biology Letters 1.4 (2005): 504-507]. The particular challenge with understanding the dynamics of this type of learning is the complex interplay of individual changes: an individual A influences another individual B, which in turn influences others (including ultimately A) as the outcome of this interaction. Such complex interaction networks are very difficult to analyse mathematically, which is why we focus on them in isolation first in this study. Combining this with other externalities (environmental social cues) will be a necessary next step to extend this framework. 

We have added a clarification of this distinction in lines 68-76 of the revised manuscript.

  1. Line 132: Please change “she” to “the individual”, as there are also male workers in termite colonies.

The phrasing was corrected accordingly.

  1. Discussion: The paper mentions that inactivity is a “by-product” of learning, and such a statement implies this behavior does not have any adaptive value. However, in other publications, there are alternative hypotheses and non-learning mechanisms examined and discussed, and some suggest “laziness” may play a functional role. For example, the lazy workers might be a reserve labor force in the colony, they might be younger and physiologically less efficient (not necessarily related to learning, but rather due to developmental plasticity mediated by social stimuli), or they might be selfishly reproducing or developing ovaries. It would be nice to introduce and discuss the non-learning mechanisms and potential functions of inactive behavior in the manuscript.

To address the above comment, the Discussion section of the revised manuscript is now strengthened by adding a paragraph discussing potential functionalities of an inactive subpopulation as suggested by previous literature including reserve workforce, resting, or patrolling to gather information. The limitations of these hypotheses have also been briefly mentioned. These changes can be found in lines 339-359 of the revised manuscript.  

Author Response

We thank the reviewer for their detailed comments and constructive feedback, which has led to a stronger and more comprehensive revised manuscript

Two fundamental types of behavioural adaptation are applied to investigate related topics of division of labour, individual learning and social learning respectively. Several results are further provided by considering behavioural dynamics and relative simulations among various environmental settings under the social and individual learning assumptions respectively. The paper seems to be mathematically correct. The results presented are interesting, although not difficult to obtain. I think the paper might be published after the minor revision suggested by the following comments.

  1. Although the paper under review is theoretical in nature, I would suggest to include some remarks regarding the motivation/physical significance of the study and one or two examples.

We assume that the reviewer is asking for empirical evidence when they refer to physical significance. While our study is focussed on a theoretical analysis, it is fully motivated by experimentally shown behaviours. Both social and individual learning are observed in insect colonies, but their relative properties are poorly understood. We are thus interested in studying the relation between these learning modalities in a general form. However, we investigate them in the context of an empirically motivated case study, the emergence of inactive subpopulations. This is particularly relevant since it has recently been explained within a theory of social learning, so that the question whether social learning itself is the underlying cause or whether the colony dynamics could arise from individual learning. We have added a paragraph to make clear that the investigation of these leaning modes is fully motivated from empirical studies [lines 97-104].

  1. I recommend the authors to carefully check the paper such that no notational errors remain. I encourage the authors to have their paper checked for syntax error.
  2. The symbols, notations and definitions could be revised to be clearer and more consistent throughout this paper.

We thank the reviewer for the careful examination of our methods and notations. We have attempted to have a full review of the symbols and notations used to correct for any potential discrepancies or vagueness.